# Application Potential of Bacterial Volatile Organic Compounds in the Control of Root-Knot Nematodes

**DOI:** 10.3390/molecules27144355

**Published:** 2022-07-07

**Authors:** Ali Diyapoglu, Muhammet Oner, Menghsiao Meng

**Affiliations:** 1Graduate Institute of Biotechnology, National Chung Hsing University, Taichung 402, Taiwan; diyapogluali94@gmail.com; 2Department of Life Science, National Chung Hsing University, Taichung 402, Taiwan; muhammet.oner053@gmail.com

**Keywords:** volatile organic compounds (VOCs), plant-parasitic nematodes (PPNs), root-knot nematodes (RKNs), biological control agents (BCAs), *Meloidogyne* spp.

## Abstract

Plant-parasitic nematodes (PPNs) constitute the most damaging group of plant pathogens. Plant infections by root-knot nematodes (RKNs) alone could cause approximately 5% of global crop loss. Conventionally, chemical-based methods are used to control PPNs at the expense of the environment and human health. Accordingly, the development of eco-friendly and safer methods has been urged to supplement or replace chemical-based methods for the control of RKNs. Using microorganisms or their metabolites as biological control agents (BCAs) is a promising approach to controlling RKNs. Among the metabolites, volatile organic compounds (VOCs) have gained increasing attention because of their potential in the control of not only RKNs but also other plant pathogens, such as insects, fungi, and bacteria. This review discusses the biology of RKNs as well as the status of various control strategies. The discovery of VOCs emitted by bacteria from various environmental sources and their application potential as BCAs in controlling RKNs are specifically addressed.

## 1. Introduction

Plant-parasitic nematodes (PPNs) are known to be one of the greatest threats to agricultural production, causing an annual crop loss of more than USD 150 billion worldwide [1,2,3,4]. To date, over 4100 species of PPNs have been reported [5,6], and they can be classified into three groups according to their lifestyles: sedentary endoparasites, e.g., root-knot nematodes (*Meloidogyne* spp.) and cyst nematodes (*Heterodera* and *Globodera* spp.); migratory endoparasites, e.g., lesion nematodes (*Pratylenchus* spp.) and burrowing nematodes (*Radopholus* spp.); and migratory ectoparasites, e.g., *Belonolaimus* spp., *Xiphenema* spp., and *Trichodorus* spp. [7]. Among them, root-knot nematodes (RKNs) are the most important agricultural pests, infecting the roots of over 3000 plant species [8,9,10].

Although chemical nematicides are still the most effective means for the management of RKNs, withdrawal of such chemical agents from the market has been continuously urged due to safety and environmental concerns [11,12,13,14]. To respond to the increasing demand for eco-friendly and sustainable management to control RKNs, methods using live microorganisms or their metabolites have been intensively explored recently.

Nematodes in soil are exposed to a diversity of microorganisms [15], of which nematophagous bacteria and fungi represent the most promising candidates to control RKNs. Bacterial species of a range of genera, such as *Bacillus*, *Pseudomonas*, and *Pasteuria*, were observed to exhibit antagonistic activity against RKNs, while the fungi that were detrimental to RKNs were commonly isolated from the phylum *Ascomycota*, *Basidiomycota*, *Zygomycota*, and *Chytridiomycota* [7,15,16,17,18]. With regard to microbial metabolites, volatile organic compounds (VOCs) have attracted research attention in recent years due to their efficacy in killing RKNs [7,19,20]. Additionally, the application of VOCs in agricultural practice could be both economically affordable and less toxic to humans than conventional nematicides [21].

This review paper summarizes (i) the general knowledge of the life cycle and genome of RKNs, (ii) the current status of the management strategies used in the control of RKNs, and (iii) recent progress in the identification of bacterial VOCs and their application potential in the control of RKNs.

## 2. Root-Knot Nematodes (RKNs)

Many economically important crops are hosts of RKNs, including tomato, potato, corn, soybean, maize, oats, wheat, and cotton [22,23,24]. The economic loss caused by RKNs has been estimated at USD 78 billion annually worldwide, accounting for half of the total loss due to PPNs [25]. Although the genus *Meloidogyne* consists of about 100 species [26], *M. incognita*, *M. arenaria*, *M. javanica*, and *M. hapla* are the four major species that infect more than 2000 plant species, particularly underground plant organs [22,27,28,29].

### 2.1. Life Cycle

The life span of RKNs is about three to six weeks with a cycle comprising embryo, juvenile (J1, J2, J3, and J4), and adult stages [22]. RKNs reproduce via diverse mechanisms but mostly by parthenogenesis. The eggs of RKNs are laid in gelatinous masses in the soil or plant residues. The worms hatch as second-stage juveniles (J2), and they immediately move toward the roots of plant hosts, attack the elongation zone, and migrate to the root tip [5,30]. When they reach the apical meristem region, they transmigrate to the developing vascular cylinder, triggering the formation of giant cells, which serve as nutrient sinks to support the growth of the nematode. The juveniles then become sedentary and undergo three more molts before they turn into adults [31,32]. In the adult stage, the worm-shaped males move out of the plant root, but the sedentary females continuously develop into pear-shaped females. Afterward, the female adults begin laying eggs (more than 1000 eggs per female) on the external surface of the root [22,33,34].

### 2.2. Genome

The whole genome of mitotic obligate parthenogenetic *M. incognita* was determined to be approximately 86 Mb, which contains 19,212 protein-coding genes, while that of meiotic facultative parthenogenetic *M. hapla* was about 54 Mb, containing 14,700 protein-coding genes [1,35]. Lately, the gene numbers of *M. arenaria*, *M. javanica*, and *M. incognita* were predicted to be 30,308, 26,917, and 24,714, respectively [36]. These genomes share some common features but with their own characteristics. One of the features shared by *M. incognita* and *M. hapla* is the possession of genes encoding distinct plant-cell-wall-degrading enzymes. A phylogenetic analysis suggested that these genes, which are absent in animals, were probably obtained via horizontal gene transfer from fungi or bacteria [37]. Since these enzymes are also present in some other PPNs of the order *Tylenchida*, the acquisition of these genes might occur earlier in an ancestor of *Tylenchida* during evolution, which supported the progress of their capability to parasitize plants [32,38,39].

The most notable differences between *M. incognita* and *M. hapla* are their genome structure and reproduction mode. *M. hapla* has an ordinary genome structure of diploid sexual species, while *M. incognita* is a hypotriploid with a proportion of one genome present in a second copy. Furthermore, *M. hapla* reproduces with meiosis, whereas *M. incognita* reproduces without meiosis and fusion of gametes.

## 3. Control Strategies for RKNs

Given the great damage to crop production due to the infestation of PPNs, a variety of methods have been used to control nematodes. These methods can be categorized into physical, chemical, and biological control strategies.

### 3.1. Physical Control Strategies

Nematodes are highly vulnerable before they penetrate the host plant’s roots. Therefore, targeting PPNs at their vulnerable stages could be effective. For instance, increasing soil temperatures above 40 °C by solarization is an effective way to reduce the number of nematodes in soil [24]. Moisture is another critical factor for the survival of nematodes. It has been highlighted that an insufficient amount of water in the soil would affect nematodes’ ability to move toward their host roots [40,41]. Flooding represents an opposite strategy to control nematodes in soils. Many PPNs are intolerant to oxygen starvation; therefore, flooding can kill nematodes by limiting their supply of oxygen. Similar effects were observed when nematodes were stored in deep water in a laboratory. To be effective in the field, the duration of anaerobiosis must be long enough to kill the nematodes. However, flooding may not be practicable for every agricultural practice. Taking into consideration the threat of global climate change, flooding would not be a good option to control PPNs. In brief, physical control strategies are less effective than conventional chemical control strategies, although the cost of physical control strategies is relatively lower [42].

### 3.2. Chemical Control Strategies

Using synthetic chemicals with the features of fumigants or nematicides to control PPNs was a common method applied in agriculture in the previous half-century [43,44]. For example, methyl bromide and dibromochloropropane were intensively used as soil fumigants due to their effectiveness. However, they are highly toxic chemicals causing acute respiratory toxicity and neurotoxicity via inhalation [45,46,47,48]. Exposure to the dibromochloropropane that had accumulated in the soil was found to influence men’s fertility and was linked to certain human cancers [49,50,51]. Therefore, its use in agriculture was banned in 1979. In addition, methyl bromide is a strong ozone-depleting substance. The use of methyl bromide in fumigation was banned globally after 2015 under the directive of the Montreal Protocol, except for quarantine and pre-shipment treatments [52,53]. Recently, a couple of less environmentally toxic chemicals have been suggested as alternatives to methyl bromide [42,54,55,56]. However, they have not yet been registered for use in agriculture [14,16]. Nevertheless, farmers need more reliable, eco-friendly, and low-cost approaches for sustainable agriculture.

### 3.3. Biological Control Strategies

Biological control refers to the suppression of a pest population, or the pest’s harmful impact, by using living organisms (natural enemies) or their metabolites [57,58]. Because biological control imitates the competition among species in nature, it is generally thought to be more environmentally friendly than chemical control. The strategies of biological control can be classified into conservation, importation, and augmentation according to the source of the deployed organisms [59]. The conservation strategy is carried out to maintain the existing natural enemies in an environment; the importation strategy is carried out to introduce exotic enemies of the pests where they do not occur naturally; and the augmentation strategy is carried out to release reared natural enemies periodically into the habitat where the pests occur [60,61].

An organism (or its metabolites) that reduces the density of the pest population is defined as a biological control agent (BCA). An ideal BCA should exert its effects by multiple mechanisms without producing harmful substances to humans and the environment [62]. Bacteria from a wide range of genera have demonstrated the capability to control RKNs [63,64]. The common genera include *Achromobacter*, *Arthrobacter*, *Bacillus*, *Burkholderia*, *Pasteuria*, *Pseudomonas*, *Rhizobium*, and *Serratia*. The beneficial effects come from mechanisms such as parasitism, niche competition, the induction of plant systemic resistance, and the production of antagonistic substances (antibiotics, toxins, enzymes, VOCs, etc.) [15,63,65].

There is a growing interest in using *Bacillus* spp. to control PPNs. For example, *Bacillus subtilis* conferred induced systemic resistance to *M. incognita* on tomato plants under greenhouse conditions [66]. The treatment of tomato seeds with several strains of *B. subtilis* as well as the cell-free supernatant reduced the number of galls and egg masses of *M. incognita*. 9H-purine, uracil, and dihydrouracil, produced by *Bacillus cereus* and *B. subtilis*, showed nematicidal activity against *Meloidogyne exigua* [67]. The activity of dihydrouracil was even stronger than that of the commercial nematicide carbofuran. *Bacillus firmus* DS-1 had nematicidal activity against *M. incognita*. The serine protease produced by this strain, known as Sep1, was toxic to both *M. incognita* and *C. elegans*. In vitro experiments on *C. elegans* demonstrated that Sep1 has a destructive effect on multiple intestinal and cuticle-associated proteins, resulting in impaired physical barriers of the worm [17]. *Bacillus amyloliquefaciens* D1 efficiently influenced the mortality of *M. incognita* J2s and suppressed its egg hatching rate; it also had a plant growth-promoting effect. *B. amyloliquefaciens* Y1 produced cyclo(d-Pro-l-Leu) that functions as a nematicide against *M. incognita* [65]. Treatments of potato plants with a recombinant *B. subtilis* strain, which secreted plant-defense elicitor peptide StPep1, effectively reduced root galling caused by *Meloidogyne chitwoodi* [68]. The treatment of cucumber and tomato plants with *Bacillus velezensis* BZR 86 significantly reduced the development of root-knot disease caused by *M. incognita*, and, as a result, the growth of the plants was enhanced [69].

*Bacillus thuringiensis* (Bt) is a spore-forming bacterium that produces parasporal crystals (Cry) during the sporulation phase. Indeed, Cry proteins have been used as biological insecticides around the world for decades. Ingestion of Cry proteins is a prerequisite for the proteins to damage the guts of insects [70]. Although most Cry proteins are toxic to insects, experiments on different nematode species have confirmed that several families of Cry proteins, including Cry5, Cry6, Cry12, Cry13, Cry14, Cry21, and Cry55, target nematodes and exhibit nematicidal activity [71,72,73]. Feeding *M. incognita* with transgenic tomato roots that expressed Cry6A decreased the reproduction rate of the worm by a factor of 4 [74]. Cry6Aa2 not only showed toxicity to J2s but also suppressed the egg-hatching rate of *M. hapla*. In addition, a pot experiment indicated that soil drenching with a mixture of spores and Cry6Aa2 could reduce the number of galls and egg masses on plant roots as well as enhancing the growth of the plant [75]. Cry5 produced by Bt strain Sbt003 adversely affected the life span and reproduction of *C. elegans*; in addition, it had a detrimental impact on the worm’s intestine [76].

*Pasteuria penetrans* is a Gram-positive nematode-parasitic bacterium. The capability of *P. penetrans* to control RKNs has been investigated in several studies. The bacterial parasitism starts when endospores of *P. penetrans* attach to the cuticle of J2 nematodes; consequently, the infected J2s show a reduction in mobility and the ability to enter the roots of plant hosts [77]. The treatment of cucumber with *P. penetrans* in greenhouse trials reduced *M. incognita* populations in the roots of the plant [78]. The RNAi-mediated silencing of the selenium-binding protein Mi-SeBP-1 of *M. incognita* increased the attachment of *P. penetrans* endospore onto the J2s’ cuticles, revealing the involvement of Mi-SeBP-1 in the adhesion of the bacterial endospore on the nematode cuticle [79].

*Pseudomonas simiae* sMB751 and its secreted cyclic dipeptide, cyclo(l-Pro-l-Leu), displayed significant nematicidal activity against *M. incognita* J2s. In fact, it was observed in a pot experiment that the fermentation broth of *P. simiae* MB751 could suppress *M. incognita* infection and confer induced systemic resistance against nematodes on tomato plants [80]. The *E. coli*-expressed and purified Nif3-family protein YqfO03, originally from *Pseudomonas syringae* MB03, had nematicidal activity against both *C. elegans* and *M. incognita* [64]. The treatment of *M. incognita*-infected bell pepper plants with *Burkholderia cepacia* Bc-2 and Bc-F strains showed a reduction in the numbers of eggs and J2s of the worm [81]. Prodigiosin, the red pigment produced by *Serratia marcescens*, had toxicity against juveniles of *M. javanica* and *Radopholus similis* [82].

## 4. Volatile Organic Compounds (VOCs)

VOCs are carbon-based, low-molecular-weight compounds that have high vapor pressure and easily evaporate at room temperature [83,84,85]. VOCs emitted by microorganisms are capable of controlling plant-parasitic fungi, insects, bacteria, and nematodes [86]. Therefore, microbial VOCs are suitable to apply to different agricultural systems with relatively low concentrations compared to agrochemicals, and supplemental spray or drench is not essential for the application of VOCs [62,87,88,89]. Microbial VOCs are diverse in terms of their chemical structures. They can be alcohols, ketones, hydrocarbons, terpenes, fatty acids, or heteroatom-containing compounds [90]. A vast number of microbial VOCs are archived in the mVOC 2.0 database, in which more than 2000 VOCs from approximately 1000 different microorganisms are categorized based on chemical structures, mass spectra, and microbial emitters [91,92].

Solid-phase micro-extraction (SPME) is widely used for the collection of VOCs. In this method, VOCs are adsorbed by the SPME fiber from the headspace of a culture medium. The adsorbed compounds are then separated with gas chromatography and further identified with mass spectrometry. The culture conditions (medium composition, oxygen level, temperature, etc.) and physiological stages of microorganisms may influence the production of VOCs in terms of chemical types and amounts [93]. For instance, *Lysobacter* strains grown on potato dextrose agar (PDA) and nutrient agar (NA) produced different VOCs. Pyrazines, decanal, pyrrole, δ-hexalactone, and ethanol were emitted as VOCs when *Lysobacter* strains were cultivated on NA; however, indole and acetoin were the major VOCs when the bacteria were cultivated on PDA [94]. A recent study showed that *B. gladioli* BBB-01 emitted dimethyl disulfide as the primary VOC when the bacterium was cultivated on LB agar, whereas 2,5-dimethylfuran was emitted when the bacterium was cultivated on PDA [92]. Although there is insufficient information on the mechanism of VOC emission, it has been reported that the production of certain bacterial VOCs is regulated by the GacS/GacA two-component regulatory system [95].

### 4.1. Biocontrol of RKNs with Bacterial VOCs

The toxicity of microbial VOCs to RKNs has been shown in numerous reports. A VOC could affect nematodes by acting as a contact nematicide, fumigant, repellent, or attractant. It could also suppress the hatching of eggs. Some of these reports are briefly described in the following text. The frequently discovered VOCs and their reported functions are summarized in Table 1.

The nematicidal activity of *Bacillus* spp. has been shown in many reports. VOCs emitted by *Bacillus megaterium* YFM3.25 inhibited the hatching of eggs and reduced the infection of *M. incognita* in a pot experiment. Among the 17 VOCs, 2-nonanone, 2-undecanone, decanal, dimethyl disulfide, and benzeneacetaldehyde accounted for the fumigant toxicity against juveniles and eggs of the worm [96]. *Bacillus atrophaeus* GBSC56 emitted methyl isovalerate, 2-undecanone, and dimethyl disulfide, which exhibited strong nematicidal activity against *M. incognita* [21]. *B. cereus* Bc-cm103 exhibited repellent activity to J2s of *M. incognita*. In addition, VOCs from Bc-cm103, mainly consisting of dimethyl disulfide and S-methyl ester butanethioic acid, displayed fumigant toxicity to *M. incognita* J2s and reduced the number of root galls on a cucumber plant in a double-layered pot test [97]. *Bacillus aryabhattai* MCCC 1K02966 emitted dimethyl disulfide, methyl thioacetate, 1-butanol, and pentane. Among the four VOCs, methyl thioacetate displayed the strongest contact and fumigant toxicity as well as repellent activity against *M. incognita* [16]. *Bacillus altitudinis* AMCC 1040 emitted eight VOCs. Of these, acetic acid, octanoic acid, 2-methyl-butanoic acid, 3-methyl-butanoic acid, 2,3-butanedione, and 2-isopropoxy ethylamine had nematicidal activity against *M. incognita* [98].

**Table 1 molecules-27-04355-t001:** In vitro activity of bacterial VOCs on *Meloidogyne incognita*.

VOC	Emitter	Effects on J2s				Egg Hatching Suppression
		Contact Toxicity	Fumigant Activity			
			Fatal	Attractant	Repellent	
Acetaldehyde	*Virgibacillus dokdonensis* MCCC 1A00493 [99]	[99]	[99]	[99]		[99]
Acetic acid	*Bacillus altitudinis* AMCC 1040 [98]	[98]				
Acetone	*Paenibacillus polymyxa* KM2501-1 [2]			[2]		
Acetophenone	*Pseudochrobactrum saccharolyticum* [100] *Arthrobacter nicotianae* [100] *Achromobacter xylosoxidans* [100]		[100]			
4-acetylbenzoic	*Paenibacillus polymyxa* KM2501-1 [2]	[2]				
Benzaldehyde	*Ochrobactrum pseudogrignonense* NC1 [101]	[101]				
Benzeneacetaldehyde	*Bacillus megaterium* YMF3.25 [96]		[96]			[96]
2,3-Butanedione	*Bacillus altitudinis* AMCC 1040 [98]	[98]				
2-butanone	*Virgibacillus dokdonensis* MCCC 1A00493 [99]				[99]	
Butyl isovalerate	*Wautersiella falsenii* [100]		[100]			
Decanal	*Bacillus megaterium* YMF3.25 [96]		[96]			[96]
2-decanol	*Paenibacillus polymyxa* KM2501-1 [2]	[2]	[2]	[2]		
2-decanone	*Paenibacillus polymyxa* KM2501-1 [2]	[2]	[2]			
Dimethyl disulfide	*Pseudochrobactrum saccharolyticum* [100] *Wautersiella falsenii* [100] *Proteus hauseri* [100] *Arthrobacter nicotianae* [100] *Achromobacter xylosoxidans* [100] *Bacillus megaterium* YMF3.25 [96] *Bacillus atrophaeus* GBSC56 [21] *Ochrobactrum pseudogrignonense* NC1 [101] *Virgibacillus dokdonensis* MCCC 1A00493 [99] *Pseudomonas putida* 1A00316 [6] *Bacillus cereus* Bc-cm103 [97] *Bacillus aryabhattai* MCCC 1K02966 [16]	[6,21,99,101]	[96,100]	[99]	[6]	[6,96]
1-(ethenyloxy)-octadecane	*Pseudomonas putida* 1A00316 [6]				[6]	[6]
Ethylbenzene	*Virgibacillus dokdonensis* MCCC 1A00493 [99]			[99]		
Ethyl 3,3-dimethylacrylate	*Pseudochrobactrum saccharolyticum* [100]		[100]			
Furfural acetone	*Paenibacillus polymyxa* KM2501-1 [2]	[2]	[2]	[2]		
(Z)-hexen-1-ol acetate	*Pseudomonas putida* 1A00316 [6]	[6]			[6]	[6]
2-Isopropoxy ethylamine	*Bacillus altitudinis* AMCC 1040 [98]	[98]				
1-methoxy-4-methylbenzene	*Wautersiella falsenii* [100] *Proteus hauseri* [100] *Achromobacter xylosoxidans* [100]		[100]			
2-Methyl-butanoic acid	*Bacillus altitudinis* AMCC 1040 [98]	[98]				
3-Methyl-butanoic acid	*Bacillus altitudinis* AMCC 1040 [98]	[98]				
Methyl isovalerate	*Bacillus atrophaeus* GBSC56 [21]	[21]				
Methyl thioacetate	*Bacillus aryabhattai* MCCC 1K02966 [16]	[16]	[16]		[16]	[16]
S-methyl thiobutyrate	*Pseudochrobactrum saccharolyticum* [100] *Wautersiella falsenii* [100] *Proteus hauseri* [100] *Arthrobacter nicotianae* [100] *Achromobacter xylosoxidans* [100]		[100]			
2-nonanol	*Paenibacillus polymyxa* KM2501-1 [2]	[2]	[2]			
2-nonanone	*Pseudochrobactrum saccharolyticum* [100] *Wautersiella falsenii* [100] *Proteus hauseri* [100] *Achromobacter xylosoxidans* [100] *Bacillus megaterium* YMF3.25 [96] *Paenibacillus polymyxa* KM2501-1 [2] *Pseudomonas putida* 1A00316 [6]	[2,6]	[96,100]		[6]	[6,96]
Octanoic acid	*Bacillus altitudinis* AMCC 1040 [98]	[98]				
2-octanone	*Pseudomonas putida* 1A00316 [6]	[6]			[6]	[6]
2-undecanol	*Paenibacillus polymyxa* KM2501-1 [2]	[2]	[2]			
2-undecanone	*Bacillus megaterium* YMF3.25 [96] *Bacillus atrophaeus* GBSC56 [21] *Pseudomonas putida* 1A00316 [6] *Paenibacillus polymyxa* KM2501-1 [2]	[2,6,21]	[2,6,96]		[2,6]	[6,96]
1-undecene	*Pseudomonas putida* 1A00316 [6]				[6]	[6]

*Paenibacillus polymyxa* KM2501-1 caused 87.6% and 82.6% mortality of *M. incognita* under both in vitro and in planta conditions, respectively. Eleven VOCs were emitted by *P. polymyxa* KM2501-1. Among them, furfural acetone and 2-decanol could attract *M. incognita* and then kill the worm by acting as fumigants or contact nematicides [2]. VOCs produced by *Virgibacillus dokdonensis* MCCC 1A00493 displayed several activities against *M. incognita*. Acetaldehyde acted as an attractant, contact nematicide, and fumigant, whereas ethylbenzene acted as an attractant and 2-butanone as a repellent [99].

*Pseudomonas putida* strain 1A00316, isolated from Antarctic soil, emitted 2-nonanone, 2-octanone, 2-undecanone, dimethyl disulfide, (Z)-hexen-1-ol acetate, 1-undecene, and 1-(ethenyloxy)-octadecane. Of these, 2-nonanone, 2-octanone, 2-undecanone, dimethyl disulfide, and (Z)-hexen-1-ol acetate showed contact nematicidal activity against *M. incognita*; however, only 2-undecanone exhibited fumigant activity. In addition, all seven VOCs suppressed egg hatching and showed repellent activity to *M. incognita* J2s in Petri plate experiments [6].

In total, 53 VOCs were identified from five bacteria, namely, *Pseudochrobactrum saccharolyticum*, *Wautersiella falsenii*, *Proteus hauseri*, *Arthrobacter nicotianae*, and *Achromobacter xylosoxidans*. Among the VOCs, S-methyl thiobutyrate, dimethyl disulfide, acetophenone, 2-nonanone, butyl isovalerate, ethyl 3,3-dimethylacrylate, and 1-methoxy-4-methylbenzene, exhibited significant nematicidal activity against both *C. elegans* and *M. incognita* in Petri plate experiments. Moreover, S-methyl thiobutyrate was the most active VOC [100]. *Ochrobactrum pseudogrignonense* NC1 significantly inhibited *M. incognita* in Petri plate and greenhouse trials. The main VOCs emitted by NC1, namely, dimethyl disulfide and benzaldehyde, also had nematicidal activity against *M. incognita* [101].

Besides *M. incognita*, some reports addressed the microbial fumigant toxicity to other *Meloidogyne* species. Three bacterial strains (*Bacillus* sp., *Paenibacillus* sp., and *Xanthomonas* sp.) emitted VOCs that were toxic to rice RKN *Meloidogyne graminicola* in both in vitro and in planta studies [102]. In vitro treatment with *P. putida*, *Microbacterium* sp., *Bacillus methylotrophicus*, and *Bacillus pumilus* caused significant mortality of *M. exigua* via the release of VOCs [103]. *Variovorax paradoxus*, *Comamonas sediminis*, *Pseudomonas soli*, *Pseudomonas koreensis*, and two strains of *Pseudomonas monteilii* were reported to exhibit nematicidal activity. They showed strong virulent effects on *M. javanica* through the production of VOCs [104].

Among the microbial VOCs identified thus far, dimethyl disulfide is the most commonly identified. In light of its toxicity to a broad spectrum of pests, dimethyl disulfide was registered by Arkema as a pesticide by the name of Paladin in 2012 [105].

### 4.2. Mechanism of Action of Bacterial VOCs

It is thought that VOCs may destroy nematodes by targeting the intestine, nervous system, surface coat, pharynx, or other tissues [2,17,106]. A recent study has claimed that VOCs cause rapid death by inducing severe oxidative stress in nematodes [21]. However, the detailed molecular mechanisms underlying the nematicidal activity of VOCs are poorly understood, with a few exceptions. A well-studied VOC, dimethyl disulfide, exerts its toxicity by blocking the activity of the enzyme cytochrome oxidase, consequently stopping the mitochondrial respiration of the pests [105].

Bacterial VOCs have also been reported to regulate the key genes involved in different signaling pathways by which plant growth is stimulated and induced systemic resistance against phytopathogens is triggered. For example, methyl isovalerate and 2-undecanone promoted plant growth and stimulated induced systemic resistance by enhancing the antioxidant enzyme activity in plant roots infested with *M. incognita* [21]. The effects of bacterial VOCs on plant morphology and physiology are discussed in a recent review paper [107].

## 5. Concluding Remarks and Future Perspectives

Driven by the concerns about the negative impacts of chemical nematicides on human health and the environment, there has been a surge of interest in the development of sustainable methods to replace the chemical strategy of controlling RKNs. A large number of reports have demonstrated that microorganisms constitute a rich source for the discovery of potentially useful VOCs in the control of RKNs. Although most of the data came from in vitro tests, some were from in planta experiments performed in greenhouse conditions. However, extensive investigations are needed to confirm whether VOCs are also effective against RKNs in open fields.

Dimethyl disulfide represents a successfully commercialized VOC, which not only is emitted by a broad spectrum of bacteria but is also effective for the control of a variety of pests. Some other VOCs, particularly the sulfur-containing ones, such as S-methyl thiobutyrate and S-methyl thioacetate, are also promising candidates because of their strong toxicity to nematodes. Further assessments of their potential in agricultural practice should be encouraged.

Investigations into how nematodes are affected by VOCs at the molecular level are still rare. Since the chemical nature of VOCs is diverse, each type of VOC might have its own mode of action. The answer to this query is not only of interest for academic research purposes but is also crucial for the development of VOCs for nematode control in the future.

## Data Availability

Not applicable.

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
