# Peer review of "Application Potential of Bacterial Volatile Organic Compounds in the Control of Root-Knot Nematodes"

_molecules, 2022, doi:10.3390/molecules27144355_

Round 1
Reviewer 1 Report
The manuscript “Application potential of bacterial volatile organic compounds in the control of root-knot nematodes" discusses the biology of root-knot nematodes as well as the status of various control strategies that researchers in the field may be interested to take a look on. Root-knot nematodes attack a wide variety of plants and can become serious pests in agriculture.
Minor comments.
Line 59. hapla instead of Hapla. In italics.
Line 194. cepacia instead of cepecia. In italics.
Author Response
Line 59. hapla instead of Hapla. In italics.
Response: “Hapla” had been changed to “hapla”. (Line 62, revised version)
Line 194. cepacia instead of cepecia. In italics.
Response: Typo corrected. (Line 210, revised version)
Reviewer 2 Report
The work presented for review concerns the possibility of reducing the population density of root-knot nematodes by bacterial volatile organic compounds. The main topic is interesting. The nematodes concerned by the work are among the most important pests of crops. Every year, these nematodes cause losses of hundreds of billions of dollars. The control options for these pests, on the other hand, are increasingly limited by international and national law. Overall, the work, although limited in scope, is well written and may be accepted for publication with minor corrections.
L74-91: The paragraph has little relevance to the topic of the work. In my opinion, it should be removed.
L107-110: In my opinion, given global climate change, flooding is not a good option to control plant parasitic nematodes.
L112: to target the life cycle of PPN - explain or correct.
L115-116: toxicological? – toxic rather.
L117-118: change “cause” to influence, carefully analyze the meaning of this sentence.
L145: remove the last sentence in the paragraph.
L157: change “increased” to influenced”.
L215-224: clearly indicate which of the listed chemical compounds are VOCs.
L247: explain “significant mortality” (10%?, 90%?).
L270: carefully analyze the meaning of this sentence, M. incognita is a nematode species.
L274: “significant death” – change to “significant mortality”.
L276: carefully analyze the meaning of this sentence, “nematicidal antagonists”???
L277: change “killing effects” to “virulent” or “pathogenic”.
L282: change 1.1 to 1.2.
L298: in vitro – italic, correct the table header formatting, remove highlighting the text in “decanal”.
L302: “A large and…” - rephrase the sentence.
Author Response
L74-91: The paragraph has little relevance to the topic of the work. In my opinion, it should be removed.
Response: Although this paragraph has little relevance, a brief introduction to the genomes of Meloidogyne species may provide an update on this progress to the readers. Therefore, we would like to ask permission from the Reviewer to keep this paragraph. (L 82-93, revised version)
L107-110: In my opinion, given global climate change, flooding is not a good option to control plant parasitic nematodes.
Response: The context is changed to “Nonetheless, flooding may not be practicable for every agricultural practice. Taking into consideration with global climate change, flooding would not be a good option to control PPNs.” (L 119-121, revised version)
L112: to target the life cycle of PPN - explain or correct.
Response: “Life cycle” was typo in this sentence. The sentences is revised as “Using synthetic chemicals with the features of fumigants or nematicides to control PPNs was a common method applied in agriculture in the previous half-century”. (L128, revised version)
L115-116: toxicological? – toxic rather.
Response: The word “toxicological” is changed to “toxic”. (L131, revised version)
L117-118: change “cause” to influence, carefully analyze the meaning of this sentence.
Response: The sentence is revised as “Exposure to the dibromochloropropane accumulated in the soil would influence men's fertility and human cancers”. (L132-133, revised version)
L145: remove the last sentence in the paragraph.
Response: The last sentence of the paragraph had been removed.
L157: change “increased” to influenced”.
Response: “increased” is changed to “influenced”. (L174, revised version)
L215-224: clearly indicate which of the listed chemical compounds are VOCs.
Response: The context is changed to “Pyrazines, decanal, pyrrole, d-hexalactone, and ethanol were emitted as VOCs when Lysobacter strains were cultivated on NA; however, indole and acetoin were the major VOCs when the bacteria were cultivated on PDA [93]. A recent study showed that B. gladioli BBB-01 emitted dimethyl disulfide as the primary VOC when the bacterium was cultivated on LB agar, whereas 2,5-dimethylfuran were emitted when the bacterium was cultivated on PDA” (L236-241, revised version)
L247: explain “significant mortality” (10%?, 90%?).
Response: The sentence is revised as “Paenibacillus polymyxa KM2501-1 caused 87.6% and 82.6% mortality of M. incognita under both in vitro and in planta conditions, respectively”. (L268-269, revised version)
L270: carefully analyze the meaning of this sentence, M. incognita is a nematode species.
Response: The sentence is revised as “Besides M. incognita, some reports addressed the microbial fumigant toxicity to other Meloidogyne species”. (L291-292, revised version)
L274: “significant death” – change to “significant mortality”.
Response: “significant death” is changed to “significant mortality”. (L296, revised version)
L276: carefully analyze the meaning of this sentence, “nematicidal antagonists”???
Response: is changed to “…were reported to exhibit nematicidal activity.” (L298, revised version)
L277: change “killing effects” to “virulent” or “pathogenic”.
Response: The sentence is revised as “They showed strong virulent effects on M. javanica through the production of VOCs”. (L299, revised version)
L282: change 1.1 to 1.2.
Response: The numbering has no error in both word and PDF formats that we originally submitted. The problem probably occurred during the submission and handling process. Anyway, the correct numbering is assured in the revised version.
L298: in vitro – italic, correct the table header formatting, remove highlighting the text in “decanal”.
Response: “in vitro” had been changed to “in vitro”. The format of the table header and “decanal” has no error in both word and PDF format of the manuscript that we submitted. The problem probably occurred during the submission and handling process. Anyway, the correct header formatting is assured in the revised version. (L321, Table 1, revised version)
L302: “A large and…” - rephrase the sentence.
Response: The sentence is revised as “A large number of reports have demonstrated that microorganisms constitute a rich source for the discovery of potentially useful VOCs in the control of RKNs”. (L329-331, revised version)

Reviewer 3 Report
This review of “Application potential of bacterial volatile organic compounds in the control of root-knot nematodes” outlines the key players in the chapters dedicated to: Root-knot nematodes (RKNs), its life cycle and genome, followed by control strategies of RKNs. These include physical, chemical and biological strategies such as bacteria from a wide range of genera with the proven capability of controlling RKNs. In the same part of the review, the authors give examples of studies dedicated to bacteria with particular focus on Bacillus thuringiensis, Pasteuria penetrans and Pseudomonas simiae. The following chapter covered volatile organic compounds (VOCs) and biocontrol of RKNs with bacterial VOCs.
The authors have summarised the most frequently discovered VOCs and their reported functions in Table 1. They have also stressed the lack of extensive knowledge and research in relation to understanding the mechanism of action of bacterial VOCs as well as the molecular details. The manuscript also indicates the directions which future research should follow, including how the nematodes are affected by VOCs. Considering that the chemical nature of VOCs is diverse, each type of VOC might have its own mode of action. Therefore, future research should address this issue, since it is crucial for the development of VOCs for nematode control in the future.
Bacterial VOCs, that are a potential alternative with positive environmental application, are reviewed in a comprehensive manner.
Other comments:
Table 1 needs extensive reformatting. The columns, as well as the headers and rows, lack a clear and readable format.
The numeration of the individual chapters needs to be verified and corrected, for instance:
2. Root-knot nematodes (RKNs)
1.1. Life cycle
1.1. Genome
2. Control strategies of RKNs
Author Response
Table 1 needs extensive reformatting. The columns, as well as the headers and rows, lack a clear and readable format.
Response: The columns, headers, and rows were re-checked. We compared the pdf file, which we submitted, with the pdf file we later downloaded from the MDPI website and we noticed that they are not the same. So that the lack of clarity in the columns, headers, and rows is only shown in the pdf file downloaded on the MDPI website but not in the submitted files. Anyway, the correct header formatting is assured in this revised version.
The numeration of the individual chapters needs to be verified and corrected, for instance:
- Root-knot nematodes (RKNs)
1.1. Life cycle
1.1. Genome
- Control strategies of RKNs
Response: The numbering was re-checked. No error was found through the submitted files (for both .pdf and .docx). Somehow only the .pdf file we downloaded from the MDPI website has such errors. Anyway, the correct numbering is assured in this revised version.